# Application of Deep Learning for Real-Time Ablation Zone Measurement in Ultrasound Imaging

**DOI:** 10.3390/cancers16091700

**Published:** 2024-04-27

**Authors:** Corinna Zimmermann, Adrian Michelmann, Yannick Daniel, Markus D. Enderle, Nermin Salkic, Walter Linzenbold

**Affiliations:** 1Erbe Elektromedizin GmbH, 72072 Tübingen, Germany; 2Faculty of Medicine, University of Tuzla, 75000 Tuzla, Bosnia and Herzegovina

**Keywords:** radiofrequency ablation, ultrasonography, artificial intelligence, image processing, computer-assisted, ablation techniques

## Abstract

**Simple Summary:**

The manual measurement of ablation zones (AZs) in radiofrequency ablation (RFA) therapy is prone to inaccuracies, highlighting the need for automated methods. Our study investigated the effectiveness of an Artificial Intelligence (AI) model, Mask2Former, in automating AZ measurements from ultrasound images, comparing its performance against manual techniques. Conducted on chicken breast and liver samples, the study found the AI model to achieve high accuracy, particularly in chicken breast tissue, with no significant difference in measurements between AI and manual methods. These results suggest that the Mask2Former model can significantly reduce variability in manual measurements, marking a step forward in the automation of AZ measurement in RFA therapy research and potentially improving the precision of treatment assessments.

**Abstract:**

Background: The accurate delineation of ablation zones (AZs) is crucial for assessing radiofrequency ablation (RFA) therapy’s efficacy. Manual measurement, the current standard, is subject to variability and potential inaccuracies. Aim: This study aims to assess the effectiveness of Artificial Intelligence (AI) in automating AZ measurements in ultrasound images and compare its accuracy with manual measurements in ultrasound images. Methods: An in vitro study was conducted using chicken breast and liver samples subjected to bipolar RFA. Ultrasound images were captured every 15 s, with the AI model Mask2Former trained for AZ segmentation. The measurements were compared across all methods, focusing on short-axis (SA) metrics. Results: We performed 308 RFA procedures, generating 7275 ultrasound images across liver and chicken breast tissues. Manual and AI measurement comparisons for ablation zone diameters revealed no significant differences, with correlation coefficients exceeding 0.96 in both tissues (*p* < 0.001). Bland–Altman plots and a Deming regression analysis demonstrated a very close alignment between AI predictions and manual measurements, with the average difference between the two methods being −0.259 and −0.243 mm, for bovine liver and chicken breast tissue, respectively. Conclusion: The study validates the Mask2Former model as a promising tool for automating AZ measurement in RFA research, offering a significant step towards reducing manual measurement variability.

## 1. Introduction

The battle against cancer remains a formidable challenge in the quest to extend the human lifespan into the 21st century [1]. In this context, minimally invasive thermal ablation techniques, such as radiofrequency ablation (RFA), microwave ablation (MWA), and high-intensity focused ultrasound (HIFU), have emerged as effective options for the treatment of tumors in organs like the liver, lung, kidney, and bone [2]. These techniques employ the targeted thermal destruction of cancerous tissues, offering a critical advantage by minimizing damage to surrounding healthy tissues [3]. 

Radiofrequency ablation has ascended as a prominent treatment modality, suitable for both curative and palliative objectives [4,5]. Its efficacy is most pronounced in tumors less than 3 cm in diameter, where a precise application of alternating current generates significant hyperthermia to induce tumor cell death [5,6]. The affected area, known as the ablation zone (AZ), is typically modeled as a three-dimensional spheroid characterized by one long axis and two short axes (SAs), which correspond to the dimensions covered by the ablation [7,8,9]. 

Despite its benefits, the success of RFA and other ablation methods hinges on the accurate assessment and monitoring of the AZ, necessitating advanced imaging techniques for real-time guidance [10]. The advent of image-guided interventions has significantly enhanced the precision of ablation therapies [11]. Although traditional imaging modalities like computed tomography (CT) and magnetic resonance imaging (MRI) provide detailed anatomical visualization, their utility is limited by the lack of real-time feedback and high operational costs [5,12,13]. Ultrasound (US) imaging, by contrast, offers a cost-effective and real-time alternative for monitoring ablation procedures [14]. Nevertheless, challenges such as image artifacts and inter-operator variability underscore the need for improvements in imaging accuracy and interpretation [5,14,15,16]. 

The integration of deep-learning algorithms with ultrasound imaging presents a promising avenue for overcoming these limitations [17,18,19,20]. By automating the recognition and measurement of the ablation zone, deep-learning (DL) models can potentially enhance the precision, reproducibility, and efficiency of thermal ablation therapies [21,22]. Convolutional neural networks (CNNs), as a particularly appropriate method of DL, were proven as a good method for object recognition and characterization [23,24,25,26]. Recent advancements in computer vision have seen a transition from CNNs to Transformer-based architectures, presenting a paradigm shift in imaging tasks. CNNs excel in image tasks but struggle with long-range dependencies vital for granular recognition. Transformer-based architectures, initially designed for natural language processing, offer a promising alternative. These models excel at processing complex spatial relationships within images and offer significant improvements in how imaging data are interpreted, providing a detailed semantic segmentation that can enhance real-time monitoring and procedural accuracy [27]. 

The objective of this study is, therefore, to explore the efficacy of employing a Transformer-based architecture for the automatic segmentation of the AZ in US images. This investigation is also aimed at facilitating the real-time monitoring of RFA progress, with the additional goal of developing an experimental setup that integrates US imaging with DL technologies. 

## 2. Materials and Methods

### 2.1. Data Acquisition and Experimental Setup

The experimental setup for this study was designed to investigate the efficacy of RFA using a bipolar RFA probe (Erbe Elektromedizin GmbH, Tübingen, Germany). The primary objective was to create ablation zones (AZs) of varying sizes to assess the performance of US imaging in conjunction with deep learning for AZ segmentation and measurement.

The RFA procedures were conducted on two types of tissue: bovine liver tissue, a commonly used surrogate for human liver in ex vivo RFA studies, and chicken breast tissue. These tissues were chosen based on their electrical properties, availability, different tissue density, and the distinct color change upon coagulation at temperatures above 60 °C, facilitating visual identification of the AZ [7,28,29]. Both tissue types were maintained at room temperature throughout the experiments.

A self-designed test stand that can ensure precise positioning of the RFA probe and the US transducer was developed (Figure 1). The RFA probe was horizontally inserted into the tissue, while the US transducer, attached to a handheld wireless linear US scanner (L7HD, Clarius Mobile Health Corp., Vancouver, BC, Canada), was positioned perpendicularly above the probe. This setup allowed for uniform imaging conditions and minimized operator-induced variability.

We used five distinct RFA durations (30, 120, 300, 600, and 900 s) to generate AZs of different sizes. The “muscle setting” on the Clarius scanner was used, and the imaging depth was set to 3 cm to ensure a consistent scale across all images. US images were captured automatically every 15 s during RFA activation, providing a detailed temporal record of the AZ development.

### 2.2. Image Processing, Deep-Learning Model, and Analysis

In the essential phase of image processing and analysis, this study employed an advanced deep-learning architecture, specifically the Mask2Former model, to analyze ultrasound (US) images captured during radiofrequency ablation (RFA) procedures. Mask2Former was chosen because of its high accuracy for ultrasound imaging processing compared to other architectures like Mask R-CNN or SOLO [30]. The initial step involved adopting the foundational code of the Mask2Former model to handle the segmentation of the AZ within US images. The architecture consists of an encoder–decoder structure and integrates a pixel decoder and Transformer decoder in the decoder stage. In the encoder, a Swin-B Transformer is used as a backbone, resulting in an overall model size of 107M parameters [27,31].

Given the limited number of images accessible for model training, the study leveraged transfer learning techniques, utilizing pre-trained weights from the Microsoft Common Objects in Context (MS COCO) dataset [32]. This approach was complemented by data augmentation strategies, including image cropping, vertical mirroring, and contrast adjustments, to enrich the dataset artificially, ensuring a robust training process despite the dataset’s constraints [33].

The model’s training was conducted through supervised learning, utilizing two distinct datasets of US images. Each dataset underwent meticulous annotation by an expert, ensuring accurate delineation of the AZ. Prior to training, individual normalization procedures were applied to the images within each dataset to cover their respective ranges. Separate models were trained for liver and chicken breast to maintain consistent accuracy across different tissue types. The datasets were then partitioned into training, validation, and test sets, following an 80/10/10 split, resulting in a distribution of 2397 training, 303 validations, and 299 test images for chicken breast tissue, and 3421 training, 457 validation, and 398 test images for liver tissue.

### 2.3. Short-Axis (SA) Measurement

The accurate measurement of the AZ short axis is essential for evaluating the effectiveness of RFA procedures. We employed several distinct approaches to SA measurement. Manual measurement was performed on the horizontally sectioned tissue using a caliper, but, as we were able to do so only after the RFA was completed, without the possibility for real-time serial measurement, this was dismissed as a possibility for a ground-truth standard.

Instead, we used the value of SA measured directly in each individual US image taken over the entire RFA run. These measurements were referred to as US diameters.

Finally, the approach that we developed for the purpose of this research involved analyzing US images through a Mask2Former model (AI method), which predicted the AZ’s mask. This method identified the largest horizontal span within the predicted mask as the SA. Subsequently, the pixel values obtained were converted to millimeters using the scale established by the ZenCore (ZenCore v2.7, Zeiss, Oberkochen, Germany) software, allowing for an accurate comparison with manual methods.

### 2.4. Statistical Methods

Prior to analysis, data were normalized to ensure uniformity across different measurement scales. This included converting all measurements to a common unit (millimeters) and aligning data points according to predefined RFA durations and tissue types. Descriptive statistics (mean, median, standard deviation, and range) were calculated for each measurement method across both tissue types (liver and chicken breast) to summarize the central tendency and variability of AZ dimensions. The Shapiro–Wilk test was employed to assess the normality of data distributions.

Comparisons between measurement methods were performed using paired t-tests or Wilcoxon signed-rank tests, depending on the normality of the data. The Bland–Altman analysis was utilized to assess the agreement between methods, with limits of agreement defined as mean difference ± 1.96 standard deviations. We assessed the data by using correlation and Deming regression to explore the alignment between two measuring methods. The significance of differences in slopes and intercepts was tested to assess method-specific biases.

A *p*-value of <0.05 was considered statistically significant for all tests. Statistical analyses were performed using GraphPad Prism 9.0.

## 3. Results

We performed a total of 308 RFA runs, out of which 59.7% were performed in liver tissue and the remaining 40.3% were performed in chicken breast tissue (Table 1). During these runs, we acquired a total of 7275 images, with a similar percentage split (58.7% and 41.3%) between liver and chicken breast tissue, as with the RFA runs. The training, validation, and testing sets of images were split in an 8:1:1 ratio. 

Thirty manual measurements of the ablation zone were performed at the end of the RFA run using a caliper. The average diameter (SD) was 16.3 mm (3.8 mm). Serial US manual measurements of the ablation zone in the liver tissue were made on US images, as previously described. A total of 398 measurements were performed, with a mean of 15.8 mm (5.9 mm), ranging from 1.83 to 29.2 mm. In the chicken breast tissue, 299 measurements were made, with a mean of 10.5 mm (4.2 mm), ranging from 2.2 to 19.4 mm.

### 3.1. Segmentation Performance of the AI Model

To evaluate the AI model’s performance, four pixel-based metrics were used. The AZ exhibited different segmentation performance for different tissue types (Table 2).

Notably, the AZ in chicken breast tissue appeared less hyperechoic in US images compared to liver tissue, as depicted in Figure 2. Conversely, the growth of the AZ around the RFA probe was shaped as an oval in chicken breast tissue, whereas, in liver tissue, the AZ assumed a more elliptical shape. As RFA progressed, increased image artifacts arose due to tissue heating and the forming of gas bubbles, thus creating a posterior shadowing effect, and complicating the assessment of the lower AZ contour. Figure 2 illustrates examples of these image artifacts.

### 3.2. Comparison of AI vs. US Manual Measurements

To begin our analysis, we compared the mean values of AI and US manual measurements in both liver and chicken breast tissue. The average diameter for AI in liver tissue was 15.6 mm (5.9 mm), while, for US, it was 15.8 mm (5.9 mm). This difference was not significant (*t*-test; *p* = 0.54). In chicken breast tissue, the mean predicted diameter for AI was 10.2 mm (4.1 mm), while the mean for US manual measurements was 10.5 mm (4.2 mm). This difference was also not significant (*p* = 0.48). Figure 3 presents a graphical comparison of the average values obtained from both methods in chicken and liver tissue.

Correlation coefficients were calculated between the diameters measured by AI and US. In bovine liver, the Pearson’s correlation coefficient was 0.965 (95% CI: 0.957–0.971; *p* < 0.001), with R^2^ = 0.931, while, in chicken breast tissue, it was 0.962 (95% CI: 0.953–0.970), with R^2^ = 0.925.

To test the alignment in measurements between the AI and US manual mode, we used a Bland–Altman plot (Figure 4). The average difference between the two methods in liver tissue was −0.259 mm (95% CI: −0.414 to −0.104), with the lower boundary at −3.339 mm and the upper boundary at 2.820 mm. In chicken breast tissue, the average difference between the two methods was −0.243 mm (95% CI: −0.374 to −0.112 mm), with the lower boundary at −2.771 mm and the upper boundary at 1.777 mm.

Finally, we utilized Deming regression to compare the AI-predicted diameter of the ablation zone in bovine liver against the US-measured diameter as the ground truth. The analysis yielded the regression equation, AI diameter = 0.9953 × US diameter + 0.3332. The equation slope of 0.9953 indicates a near one-to-one relationship between the AI predictions and the US measurements, thus demonstrating the AI model’s accuracy (with a small underestimation) in assessing the diameter of the ablation zone. The Y-intercept was found to be 0.3332, suggesting a small systematic bias in the AI predictions. 

The same analysis was performed for chicken breast tissue and the resultant regression formula, AI diameter = 1.014 × US diameter + 0.09540, signifies a slope of 1.014. Here, the AI model performed with a slight overestimation. The Y-intercept, determined to be 0.09540, hints at a minor systematic bias in the AI predictions.

Both final analyses underline the precision of the AZ diameter prediction by our AI algorithm (Figure 4).

## 4. Discussion

This study explores the potential of automated AZ measurements in US images through the application of a Mask2Former model, aiming to enhance both laboratory processes and the outcomes of RFA research. The model underwent supervised training to identify and delineate AZs in US images captured at 15 s intervals during bipolar RFA activations, focusing separately on chicken and liver samples. A custom-built test stand facilitated straightforward and consistent laboratory procedures. AZ labelling within these images was conducted by a single expert to train the Mask2Former model. Short-axis measurements were conducted to compare the performance of the AI-predicted diameter in comparison to the manually measured US diameter.

Machine learning (ML) has found extensive applications within the medical sector, notably in image analysis and diagnostics [34,35,36]. Different AI models trained via supervised learning are frequently employed for their capabilities in US image tasks [14,25,36]. Nonetheless, a significant challenge is the limited amount of available training samples, potentially hindering the effectiveness of AI training [17,25]. To address this issue, data augmentation strategies were employed to enrich the dataset and prevent model overfitting, a practice supported by various studies. This step was essential in enhancing the model’s ability to accurately recognize and delineate the AZ across different tissue types [36,37]. 

The manual measurement, serving as the primary benchmark against which other methods are evaluated, is a staple in laboratory practices. It relies on the tissue color change upon coagulation, an indicator of histopathological damage from thermal exposure [38,39]. However, we decided against using it, primarily as it does not allow for obtaining a large number of dynamic measurements needed for accurate comparison and for the real-time tracking of the size of ablation zone, which is one of the major clinical requirements in practical use.

Additional factors influencing manual measurements include the specifics of tissue sectioning and inherent tissue characteristics, such as the presence of vessels or muscle fibers, which may distort the AZ. Accurate slicing at the RFA probe’s height is crucial, as the AZ typically forms spherically around it [7]. This requirement underscores the potential advantages of automated measurement techniques, which promise consistent and precise SA measurements in undisturbed tissue, mitigating the impact of human variance. AI-enhanced image segmentation in conjunction with US imaging emerges as a promising approach to address these limitations [5,25,33].

Ultrasound imaging is a cornerstone of diagnostic and therapeutic procedures, including image-guided thermal ablation, prized for its real-time imaging capabilities [5,12]. However, US’s grayscale imaging limits lesion clarity compared to CT or MRI, intensified by challenges in tissue visibility and the effects of hyperechoic structures and gas. Despite these limitations, the US remains preferable for routine laboratory use over CT and MRI, given the latter’s extensive technical and material requirements [12,13]. 

The efficacy and reliability of the SA measurement via US have been substantiated across studies, though outcomes can vary based on equipment brand, scanner orientation, and distance from the transducer to the target [40,41,42,43]. The Clarius US scanner, a modern handheld, wireless device, demonstrates no significant compromise in image quality, maintaining measurement accuracy on par with traditional stationary systems [44].

The main goal of the present study was to compare the AI-predicted diameter against the ground truth—in our study, the manually measured US diameter. AI-derived SA measurements were extracted from the largest horizontal span across the predicted mask within the US image. This approach was grounded on the assumption that the AZ uniformly encircles the RFA probe’s center, although deviations caused by nearby vascular structures or other anomalies could challenge this assumption [8]. Given that AI utilized the same image set as the US method, it was subject to similar challenges of image artifacts, which could lead to overestimations of the SA in comparison to manual measurements. This is consistent with the broader literature noting the complexities of interpreting US images with AI due to image artifacts [16,45]. It is important to emphasize that overestimation of the SA size regularly occurs with US imaging; therefore, all methods that rely on US will have a certain amount of overestimation, which needs to be taken into account, especially in future research in real-life clinical settings.

When examining AI’s performance relative to manual measurements, it was noted that AI more accurately delineated the AZ in chicken breast tissue compared to liver tissue (Figure 2). This discrepancy may be attributed to the differential impact of US image artifacts on tissue visibility, with chicken breast tissue presenting fewer disturbances. This observation is supported by comparisons with existing literature, where AI’s efficacy in AZ detection showcases the potential for more accurate assessments in conditions with less ultrasonic interference [46]. The challenges associated with ultrasound image artifacts, such as shadowing and speckle noise, were minimized by allowing the model to learn the representation of such artifacts during the training process. As the model was trained with the expert’s labels, it behaves in a similar fashion regarding the AZ.

The results gained with Bland–Altman plots and Deming´s regression demonstrate an excellent alignment of the AI-predicted diameter with manual measurements, with negligible systematic bias and a very slight propensity for underestimation. However, it appears that this alignment is more variable in small AZs, notably, AZ < 10 mm in liver tissue and AZ < 5 mm in chicken breast tissue (Figure 4). Conversely, AI demonstrated a higher accuracy for larger SAs, especially in chicken breast tissue, likely due to the high acoustic impedance of chicken breast tissue affecting AZ visibility [47,48,49]. A smaller AZ corresponds with the start of RFA, where the acoustic properties of tissue are prone to the huge variability due to uneven heating of the tissue or steam formation next to the RFA electrodes which may at least partially explain this phenomenon. For future research into the clinical use of our method, it is important to consider that AI-predicted measurements of AZ at the beginning of the ablation may be less reliable.

It is important to mention several limitations of the present study, which primarily stem from its in vitro nature, affecting the generalizability of the findings to clinical settings. Other potential limitations include the controlled environment not fully replicating the complexity of live tissue characteristics and responses to RFA, and the use of a specific set of tissues (chicken breast and liver) which may not represent the diversity found in human pathology. Another limitation is the fact that only homogenous imaging conditions were present, without movement artifacts or blood flow. Additionally, the study’s focus on a single AI model (Mask2Former) limits exploration of alternative or potentially more effective AI approaches. Yet, the clear potential of our approach proves its applicability in laboratory conditions but warrants further investigations regarding the clinical applicability.

The evaluation of clinical applicability is, of course, a further necessary step in the exploration of this field, particularly in terms of assessing its clinical potential for both percutaneous ultrasound-guided (i.e., in the liver) and endoscopic ultrasound-guided (i.e., in the lung and pancreas) RFA procedures. Based on the results of the present study, we strongly believe that this approach offers clear potential for the reliable real-time tracking of ablation zone size in the clinical setting.

## 5. Conclusions

In conclusion, our study demonstrated the potential of using a Mask2Former model for the automatic delineation and measurement of ablation zones in ultrasound images, offering a promising tool for enhancing the accuracy and efficiency of radiofrequency ablation research. While the AI model showed notable precision compared to manual methods, variations across tissue types and the variability with smaller ablation zones highlight areas for further development. Future work should focus on refining the AI model’s adaptability and accuracy to fully leverage AI in clinical RFA applications. This will improve patient outcomes by optimizing therapeutic interventions and transferring and using the model for the real-time monitoring of US-guided RFA ablation in clinical settings.

## Figures and Tables

**Figure 1 cancers-16-01700-f001:**
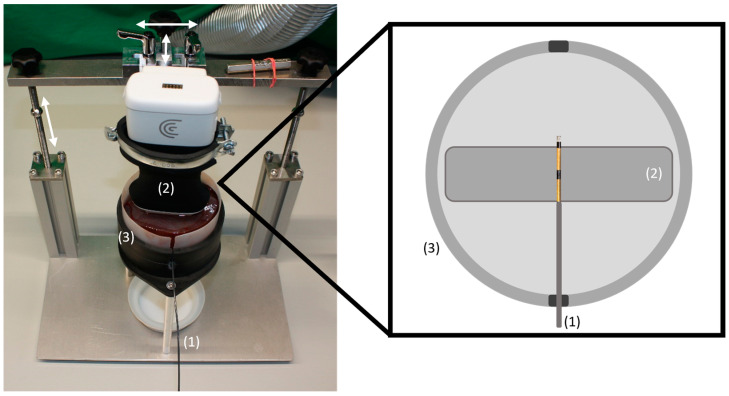
Self-designed test stand with liver tissue. The RFA probe (1) was horizontally introduced in the tissue which was placed in the tissue cup (3). The US transducer (2) was placed horizontally and perpendicular to the RFA probe at the location of the separator, as shown in the schematic on the right. For the adjustment the US transducer could be moved in three dimensions indicated by the white arrows.

**Figure 2 cancers-16-01700-f002:**
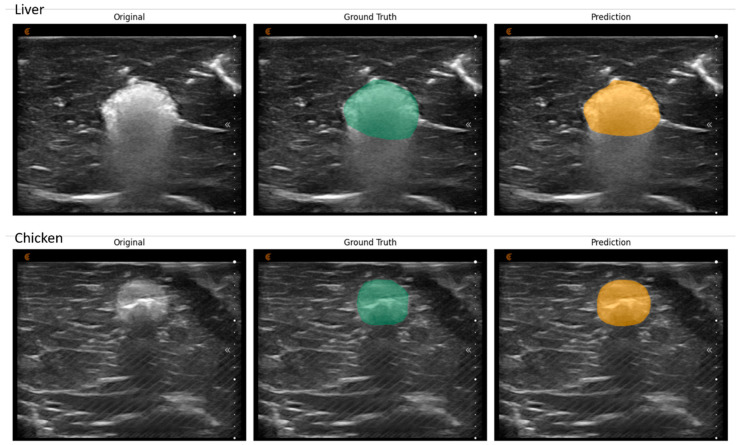
Representation of AZ in US images of liver (first row) and chicken tissue (second row) and the labelled mask (green) and predicted mask (orange). The AZ is less hyperechoic represented in chicken breast tissue as in liver tissue. Larger image artifacts can be observed underneath the AZ in liver tissue with ongoing RFA, making the assessment of the under AZ contour difficult. An acoustic shadow can be observed in the case of chicken breast.

**Figure 3 cancers-16-01700-f003:**
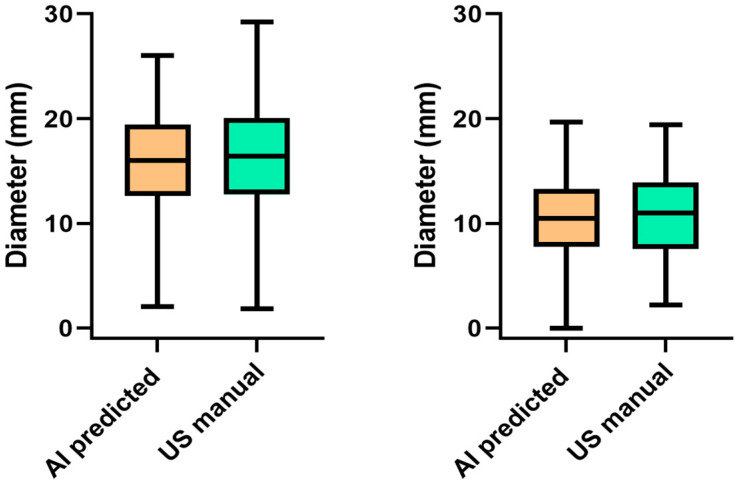
Box-plot diagrams comparing the mean values of AI predicted and US manual measured diameter of RFA ablation zones in bovine liver tissue (**left**) and chicken breast tissue (**right**).

**Figure 4 cancers-16-01700-f004:**
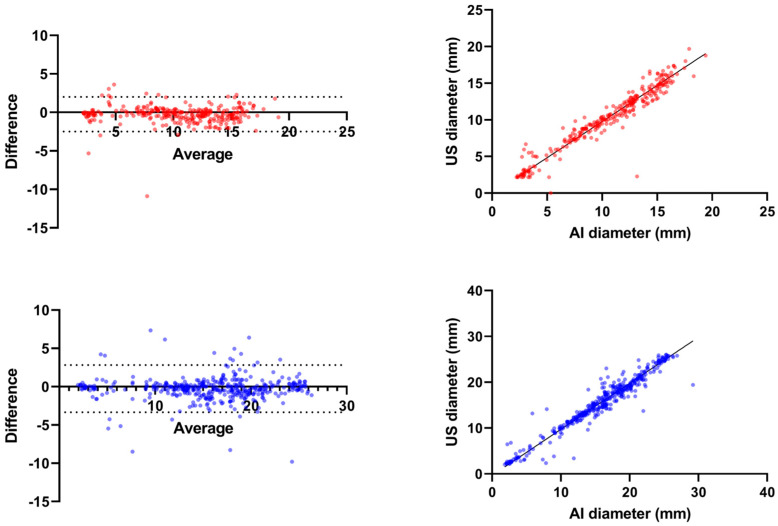
Bland - Altman plots (**left**) for chicken breast tissue (red) and bovine liver (blue) depicting the reliability of AI predicted diameter in comparison with US manual measurement as ground truth. Deming regression plots (**right**) for chicken breast tissue (red) and bovine liver (blue), demonstrating the reliability of AI−predicted diameter in comparison with US manual measurement as ground truth.

**Table 1 cancers-16-01700-t001:** Baseline characteristics of number of RFA and sample split for AI training.

	Total Number of RFA Runs	Training Set	Validation Set	Test Set	Total Number of US Images
Liver	184	147	20	17	4276
Chicken	124	98	15	11	2999

**Table 2 cancers-16-01700-t002:** Evaluation metrics of the trained Mask2Former model according to the tissue type. Higher metrics were scored for chicken breast tissue. The amount of test images used for each tissue type is given by *n*.

Tissue Type	*n*	Accuracy [%]	Sensitivity [%]	Specificity [%]	F1-Score [%]
Liver	398	98.5	88.8	99.3	89.7
Chicken breast	299	99.4	91.9	99.7	92.6

## Data Availability

The data presented in this study are available upon request from the corresponding author due to the proprietary algorithms created.

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
