# Peer review of "Application of Deep Learning for Real-Time Ablation Zone Measurement in Ultrasound Imaging"

_cancers, 2024, doi:10.3390/cancers16091700_

Round 1

Reviewer 1 Report

Comments and Suggestions for Authors

My few suggestions are listed below.

1. Introduction needs to be detailed.

2. Please write your contributions at the end of Section 1 in bullet form.

3. In Section 2, a pseudo code should be placed for readers' better understanding.

4. Title and summary are ok.

5. Results and findings are fine and discussion placed therein gives a good insight of the research done.

Author Response

Dear Reviewer.

Thank you for your valuable comments and suggestions.
Please find below and attached as word file our point-by-point response.

  1. Introduction needs to be detailed.

A: We have rephrased and re-written the introduction where necessary to make it more readable.

  1. Please write your contributions at the end of Section 1 in bullet form.

A: The contributions of all authors are listed according to the journals policy. If desired or also requested by the editors, we will change the format and write them in bullet form.

  1. In Section 2, a pseudo code should be placed for readers' better understanding.

A: We have created a pseudo code (see attached document) but felt that including the code as a separate image or table would make the manuscript too long. However, if you and the editor think it would be worth including the pseudo code, we will be happy to do so.

  1. Title and summary are ok.

A: Thank you for your comment. We appreciate this.

  1. Results and findings are fine and discussion placed therein gives a good insight of the research done.

A: Thank you for your comment. We appreciate this.

Reviewer 2 Report

Comments and Suggestions for Authors

This paper Highlighting the potential of artificial intelligence in enhancing the precision and efficiency of radiofrequency ablation research, suggesting improvements in real-time monitoring and measurement of AZ in ultrasound imaging, which could lead to more effective treatments.

1. How does the Mask2Former model handle the challenges associated with ultrasound image artifacts, such as shadowing and speckle noise, which can significantly impact the accuracy of ablation zone measurements?

2. Considering the variable echogenicity between different tissues, what modifications or adaptations were made to the Mask2Former model to ensure consistent accuracy across different tissue types, such as liver and chicken breast tissue?

3. A more detailed discussion on the model's performance across a broader range of tissue types and under various imaging conditions could offer insights into its generalizability and limitations.

4. Comparison with other deep learning architectures or models could help contextualize the performance of Mask2Former within the current landscape of AI in medical imaging, highlighting its strengths and potential areas for improvement.

5. The writing style of this paper needs to be improved. Authors should spend some time improving it to read the paper more smoothly.

The related works section is very short and no benefits from it. I suggest increasing the number of studies and add a new discussion there to show the advantage.

Overall the work is of good quality and the obtained results are interesting, but the above points should be addressed to improve the current version and to iron out the gaps in the study for further experimentation.

Author Response

Dear Reviewer.

Thank you for your valuable comments and suggestions.
Please find below our point-by-point response.

  1. How does the Mask2Former model handle the challenges associated with ultrasound image artifacts, such as shadowing and speckle noise, which can significantly impact the accuracy of ablation zone measurements?

A: The artifacts exist throughout the datasets, therefore the model learns the representation during the training process. Since the model depends on the expert’s labels, it will behave in a similar fashion regarding AZ in shadowing area.
We have addressed this in the discussion (see lines 358-361)

  1. Considering the variable echogenicity between different tissues, what modifications or adaptations were made to the Mask2Former model to ensure consistent accuracy across different tissue types, such as liver and chicken breast tissue?

A: One version of the model was trained for each liver and chicken breast. For the respective trainings, each dataset was pre-processed with an individual normalization for the images to cover their range. We have included this into the Material and Methods section (see line 140-148)

  1. A more detailed discussion on the model's performance across a broader range of tissue types and under various imaging conditions could offer insights into its generalizability and limitations.

A: Thank you for this remark, as it is an important aspect. We have prioritized a standardized lab environment for this study to ensure a good reproducibility. For a clinical use, we still need to figure out, which factors can not be controlled for and adapt the dataset accordingly. This limitation was already included as a limitation but was revised accordingly (see line 380-381).

  1. Comparison with other deep learning architectures or models could help contextualize the performance of Mask2Former within the current landscape of AI in medical imaging, highlighting its strengths and potential areas for improvement.

A: Initial experiments with the Mask-R-CNN architecture revealed unsatisfying results and thus neglected. Since the Mask2Former architecture fit our practical needs, we did not focus on comparison of further architecture. This was also shown by Yuan Y. et al. comparing Mask2Former with MaskRCNN or SOLO for imaging lateral cervical lymph node using ultrasound. We have included this information to the manuscript (see Line 127-128).

Yuan Y, Hou S, Wu X, Wang Y, Sun Y, Yang Z, Yin S, Zhang F. Application of deep-learning to the automatic segmentation and classification of lateral lymph nodes on ultrasound images of papillary thyroid carcinoma. Asian J Surg. 2024 Mar 6:S1015-9584(24)00401-9. doi: 10.1016/j.asjsur.2024.02.140

  1. The writing style of this paper needs to be improved. Authors should spend some time improving it to read the paper more smoothly.

A: Thank you for your comment.
We have re-written and revised the manuscript and hope this revision will make it easier to read.

The related works section is very short and no benefits from it. I suggest increasing the number of studies and add a new discussion there to show the advantage.

Overall the work is of good quality and the obtained results are interesting, but the above points should be addressed to improve the current version and to iron out the gaps in the study for further experimentation.

A: Thank you for your valuable comment to improve the manuscript. We hope we could address your concerns with our changes and improve the manuscript accordingly.

Reviewer 3 Report

Comments and Suggestions for Authors

This is an interesting research that used Mask2Former model to segment the ablation zones in RF ablation therapy. It can achieve automating AZ measurement in real time and has lots of potentials for improving the treatment assessments and cost. 

Author Response

Thank you for your valuable and gentle comments. We appreciate this very much.